# Effective Quality Breeding Directions—Comparison and Conservative Analysis of Hepatic Super-Enhancers between Chinese and Western Pig Breeds

**DOI:** 10.3390/biology11111631

**Published:** 2022-11-08

**Authors:** Yi Zhang, Jinbi Zhang, Caixia Wang, Xinxin Qin, Yuge Zhang, Jingge Liu, Zengxiang Pan

**Affiliations:** 1Laboratory of Statistical Genetics and Epigenetics, College of Animal Science and Technology, Nanjing Agricultural University, Nanjing 210095, China; 2College of Animal Science and Technology, Jinling Institute of Technology, Nanjing 211169, China

**Keywords:** *cis*-regulatory elements, super-enhancers, Chinese pig breeds, Western pig breeds

## Abstract

**Simple Summary:**

In this study, we identify the promoters, typical enhancers (TEs), and super-enhancers (SEs) in the livers of Chinese local and Western commercial pig breeds. Western breeds included fewer SEs in number, while more QTLs of growth-related economic traits were associated with these SEs. Comparison among different porcine tissues and liver tissues from pigs, humans, and mice suggested a high tissue specificity of SE. We concluded that intense selection could concentrate functional SEs; thus, SEs could be applied as effective detection regions in genomic selection breeding.

**Abstract:**

The transcriptional initiation of genes is closely bound to the functions of *cis*-regulatory elements, including promoters, typical enhancers (TEs), and recently-identified super-enhancers (SEs). In this study, we identified these *cis*-regulatory elements in the livers of two Chinese (Meishan and Enshi Black) and two Western (Duroc and Large White) pig breeds using ChIP-seq data, then explored their similarities and differences. In addition, we analyzed the conservation of SEs among different tissues and species (pig, human, and mouse). We observed that SEs were more significantly enriched by transcriptional initiation regions, TF binding sites, and SNPs than other *cis*-elements. Western breeds included fewer SEs in number, while more growth-related QTLs were associated with these SEs. Additionally, the SEs were highly tissue-specific, and were conserved in the liver among humans, pigs, and mice. We concluded that intense selection could concentrate functional SEs; thus, SEs could be applied as effective detection regions in genomic selection breeding.

## 1. Introduction

Cis-regulatory elements are DNA sequences with transcriptional functions, such as promoters and enhancers. They can be recognized by sequence-specific transcription factors (TFs), which cooperatively recruit other co-factors to activate or resist transcription activity [1,2]. Lately, super-enhancers (SE) have emerged as a newly-defined powerful *cis*-regulatory element. SE comprises a cluster of typical enhancers (TEs) densely occupied by master regulators and mediators. SE is speculated to act as a switch to determine cell identity and fate [3]. With the development of chromatin immunoprecipitation techniques, histone modifications, which mediate gene expression or repression during various critical biological processes via chromatin modification [4], have been regularly associated with *cis*-regulatory elements [5]. Generally speaking, promoters of actively transcribed genes are marked by high levels of H3K4me3 [6], while enhancers can be identified by H3K27ac modification [7,8]. SEs can be identified by the H3K27ac marker, which has higher intensity, larger fragment length, and higher density of TFs and transcriptional coactivators [9,10,11]. Furthermore, SEs are enriched for single nucleotide polymorphisms (SNPs) associated with a broad spectrum of diseases [12].

The pig is undoubtedly one of the most important livestock species, serving as both a significant food source and an excellent animal model for biomedical research in humans thanks to similar physiological features and organ sizes [13,14]. Chinese and Western pig breeds have a divergent evolutionary history, and were independently selected for breeding some 9000 years ago [15]. Long-term domestication and modern breeding have resulted in genetic/epigenetic modification variation and phenotypic differences (growth, development, reproduction, and production performance) between Chinese and Western pig breeds. The Duroc and Large White are traditional Western pig breeds, both of which are subjected to intense selection for major economic traits including growth rates, muscle mass, and feed efficiency [16,17]. On the contrary, local Chinese pig breeds are subjected to fewer selection pressures and have more diverse performance traits than western breeds. For example, Meishan and Enshi Black, which have a lower growth rate, show comparatively more robust roughage tolerance, higher intramuscular fat, and higher reproductive performance [18,19]. The liver is a critical metabolic organ that governs numerous physiological processes such as macronutrient metabolism, cholesterol homeostasis, endocrine control, xenobiotic response, immune system support, and blood volume regulation. It governs numerous physiological processes, especially regarding metabolism, and drives feeding efficiency, growth, and other economical traits in pigs [20]. Therefore, the liver’s characteristics and slightly different functions could be a source of commercial traits such as feeding efficiency and disease resistance [21,22].

Cis-elements have essential roles in development, and their divergence is a common cause of evolutionary changes [23]. Therefore, the pig breeding process must involve changes in enhancer activities, which may be caused by genetic activities such as SNP substitutions and insertions and/or deletions that finally cause enhancement or repression of gene activities of the target traits. Such ideas have been observed and reported in recent studies [24,25]. In addition, a study of the evolving landscape of SEs has confirmed their gene regulatory functions during cell differentiation in multiple lineages [26]. As a more powerful *cis*-element, SE may play a more important role during the development and breeding process. However, the relationship between SE differences and phenotypes among breeds remains unknown. In this study, we hypothesize that *cis*-elements, especially SEs, change during breeding and are highly correlated with key genes of selected traits. We first identified the promoters, TEs, and SEs in the livers of two Chinese (Meishan and Enshi Black) and two Western (Duroc and Large White) pig breeds using ChIP-seq data and explored the effect of breeding on *cis*-element repertoires. Then, the SE profiles of different porcine tissues and different species (pig, human, and mouse) were generated and compared to reveal the tissue conservation and interspecific difference. The resulting study provides new insights into the regulatory features and practical breeding values of SEs.

## 2. Materials and Methods

### 2.1. Available Public Data

H3K27ac and H3K4me3 ChIP-seq data for eleven tissues (cerebellum, cerebrum, fat, heart, kidney, liver, lung, muscle, pancreas, spleen, thymus) of four pig breeds (Meishan, Enshihei, Large White, and Duroc) were downloaded from PRJNA597497 (https://www.ebi.ac.uk/ena/browser/view/PRJNA597497?show=reads, accessed on May 2022). H3K27ac ChIP-seq data of human and mouse liver tissue were downloaded from PRJEB6906 (https://www.ebi.ac.uk/ena/browser/view/PRJEB6906?show=reads, accessed on January 2015). The individual accession IDs are listed in the Appendix A.

### 2.2. ChIP-Seq Data Processing

For ChIP-seq data, the raw data were trimmed by Trim galore (https://www.bioinformatics.babraham.ac.uk/projects/trim_galore/) with default parameters in paired-end mode. The reads were generated by removing low MAP(Q) reads < 25 and unmapped reads, and duplicated using SAMTools v1.8 [27]. The ChIP-seq reads and input reads were aligned to susScr11, hg38, and mm10 genome assemblies using BWA (version 0.7.17) [28]. The filtered reads were used for downstream analyses. H3K4me3 and H3K27ac peaks were identified using MACS2 v2.2.7. (parameter settings: macs2 callpeak -t input_bam -c Control.bam --nomodel -f BAMPE --broad -g genome.size --keep-dup 1 –broad) [29]. For tissues with two replicates, their peaks overlapped at least 80% reciprocally, and were used for downstream analysis using BEDTools (version2.30.0) [30] with “intersect -f 0.8”. The Pearson correlation coefficient between each biological replication for H3K4me3 and H3K27ac was calculated using Deeptools (version 3.5.1) [31] to check repeatability between biological replicates. The reads were normalized, and heatmaps were generated using Deeptools (parameter settings: computeMatrix reference-point --referencePoint TSS -R scrofa.Sscrofa11.1.104.gtf -a 5000 -b 5000 --skipZeros -S input.bw -o plotHeatmap -m -outout.png --colorMap RdBu --whatToShow ‘plot, heatmap and colorbar’ --zMin -3 --zMax 3) (version 3.5.1).

### 2.3. Identification of cis-Regulatory Elements

The *cis*-regulatory elements, including promoters and enhancers, were identified based on the above enriched regions. The enriched regions for H3K4me3 were defined as potential promoters. 

The super-enhancers and typical enhancers were identified by H3K27ac peaks located away from TSS regions (outside 2.5 kb upstream and downstream of TSS) using the ROSE algorithm (https://bitbucket.org/young_computation/rose) with default parameters. Briefly, the TFs were ranked by normalized peak counts and thresholds above a flex-point (slope > 1) and identified as super-enhancers. *cis*-Regulatory Elements were annotated using ChIpseeker (version 1.20.0) [32] and ROSE_geneMapper.py.

### 2.4. Differential Analysis of Promoters and TEs

Differentially enriched regions (DERs) between Chinese breeds and Western breeds were identified from a consensus set of peaks using DiffBind [33] (FDR Threshold = 0.05). DERs annotated using ChIpseeker (version 1.20.0) [32].

### 2.5. Identification of Conserved and Specific SEs

The conservation and specificity of SEs were analyzed using BEDTools (version 2.26.0). This study required conserved SEs that overlapped more than 20% of the base pair unless otherwise stated. Specific and conserved SEs were identified according to the number of overlapping SEs across breeds, tissues, and species.

### 2.6. Motif Enrichment of Specific SEs

The “findMotifsGenome.pl” script from the HOMER (version 4.9.1) package was used to search for enriched motifs (parameter setting: findMotifsGenome.pl SEs.bed pig SE_motif -len 8,10,12). The whole genome was used as the background. Factors with *p*-values lower than 1 × 10^−2^ were considered enriched.

### 2.7. Comparison of Orthologous SEs and TEs

We converted SEs and TEs in susScr11 and mm10 hg38 and coordinates using UCSC LiftOver tools (https://genome.ucsc.edu/cgi-bin/hgLiftOver) with a minimum match of 0.5. The TEs and SEs that could be converted to human genomic locations were considered sequence-conserved TEs and SEs. Furthermore, TEs and SEs were considered usage conserved if the corresponding human homologous sequence was covered by TE and SE. To calculate the percentage of TE and SE sequences overlapping with genomic features, TE and SE annotations were compared to RefSeq Gene annotations using the BEDTools (version 2.26.0) intersect function.

### 2.8. Other Bioinformatic Analysis

The ChIpseeker v1.20.0 R package [32] was used to profile the distribution of SEs and TEs throughout the Sscrofa11.1 genome. GO and pathway analysis was carried out using DAVID (https://david.ncifcrf.gov). Normalized ChIP-seq data were visualized using the Integrative Genomics Viewer (IGV). To predict the SE functions, we determined SE regions and QTLs on the genome with BEDTools v2.30.0, and the porcine QTL database was downloaded from the Animal QTLdb [34] (https://www.animalgenome.org/cgi-bin/QTLdb/SS/index).

## 3. Result

### 3.1. Profiling of Promoter, TE, and SE in Liver

To identify promoter, TE, and SE, we analyzed ChIP-seq data of H3K4me3 and H3K27ac markers of liver tissues in Meishan (MS), Enshihei (ES), Large White (LW), and Duroc (DR) pigs with two biological replicates. The Pearson correlation first confirmed satisfactory repeatability between individuals (correlation coefficient between 0.93 and 0.98) (Figure 1A). The distribution of H3K4me3 and H3K27ac suggested that the H3K4me3 peaks showed a stronger enrichment in the TSS (±5 kb) regions (Figure 1B). Then, we identified an average of 28,116 promoters, 20,295 TEs, and 1248 SEs in the porcine liver, respectively. When identifying TEs and SEs using the ROSE algorithm, the regions with a slope greater than one were defined as SEs, and the remainder were defined as TEs (Figure 1C). The number of TEs showed the most variation between breeds, and Western breeds generally had more TEs than Chinese ones (Figure 1D).

The genomic features analysis revealed that the majority of promoters, TEs, and SEs were located in the proximal promoter (within 1kb of TSS) (52.80%, 38.03%, and 87.89%) (Figure 1E). Notably, the SEs covered more proximal promoter regions and contained a higher density of proximal transcription factor (TF) binding sites among the three *cis*-elements (Figure 1F). The peak signal plot showed that the signal intensity was higher across the SE regions than the TE regions (Figure 1G), in agreement with the more substantial transcriptional power of SEs.

### 3.2. Comparison of Promoter and TE Variants across Different Porcine Breeds

We used the consensus peaks of the two groups (FDR < 0.05, Diffblind) to investigate the enrichment of promoter or TE regions between Chinese and Western breeds. Both the heatmap clustering (Figure 2A) and PCA analysis (Figure 2B) suggested distinct differentially enriched regions (DERs) of H3K4me3 and H3K27ac between Chinese and Western breeds. Interestingly, the H3K4me3 DERs were highly consistent in Chinese breeds; in contrast, the H3K27ac DERs were highly consistent in Western breeds. This result might be due to entirely different breeding methods. After identifying 654 promoter DERs (105 and 549 enriched in Chinese and Western breeds) and 1491 TE DERs (802 and 689 enriched in Chinese and Western breeds) (Appendix A), we annotated genes that overlapped with these DERs and explored their biological function enrichment.

Regarding promoters, 380 and 94 genes were associated with H3K4me3 DERs of Chinese and Western breeds, and were enriched with “the response to progesterone, aging, positive regulation of the apoptotic process” and “positive regulation of transcription from RNA polymerase II promoter, response to the drug, insulin response, retinoid metabolic process”, respectively (Figure 2C). For TE, we identified 546 and 597 genes associated with H3K27ac DERs in Chinese and Western breeds, and their associated biological functions were similar to the promoter (Figure 2D, Appendix A).

### 3.3. Conservation and Variants of SEs in Different Breeds

To assess SE features in different pig breeds, we first identified 1248, 1332, 1021, and 1094 SEs in the livers of MS, ES, LW, and DR breeds, respectively (Figure 3A, Appendix A). The Venn diagram of SEs suggests that approximately 70% of SEs were conserved in all four pig breeds (Figure 3B), which we term breed-conserved SEs. A total of 1113 genes related to breed-conserved SEs were used for functional enrichment analysis. The results revealed that pathways such as P13K-Akt signaling, FoxO signaling, glucagon, and insulin resistance were the most involved (Figure 3C, Appendix A). Then, we focused on breed-specific SEs in either Chinese or Western breeds. There were 371 Chinese and 226 Western breed-specific SEs, associated with 162 and 161 genes, respectively (Appendix A). KEGG enrichment suggested that these specific genes were intensively enriched in hormone and metabolic regulation processes, such as insulin resistance, prolactin signaling, glycine/serine/threonine metabolism, the pentose phosphate pathway and carbon metabolism (Appendix A). Interestingly, we found that the Western-specific SEs were particularly associated with genes involved in growth hormone synthesis, secretion, and action pathways (Appendix A), such as the STAT family of transcription factors (*STAT5A* and *STAT5B*), the somatotropin/prolactin family of hormones (*GH1* and *PRL*) and insulin-like growth factor binding protein (*IGFBP*) (*IGFBP1*) (Figure 3D and Appendix A). Furthermore, we applied motif analysis to examine breed-specific transcription factors. In the Chinese-specific SEs, we identified significant motifs such as *X-box* and *PIT-1*, while in the Western breeds SEs the binding sites of the MYB family, including *MYB92*, *ATY13*, and *MYB17*, were highlighted (Figure 3E).

To investigate the relationship between breed-specific SEs and qualitative traits, we mapped the SEs to the QTL regions of five trait categories (meat, health, exterior, production, and reproduction) using the pig QTL database. We observed that the QTLs associated with breed-specific SEs were quite different. First, Western breed-specific SEs had much higher global QTL overlapping rates, especially in meat quality-related traits (Figure 3F). Second, when looking into the detailed traits, Western ones were associated with shoulder subcutaneous fat thickness, backfat at the rump, coping behavior, average daily gain, and loin muscle area traits. In contrast, Chinese ones mainly overlapped with head weight, average daily gain, drip loss, teat number, and LDL cholesterol (Figure 3G, Appendix A). These results provide clear evidence of long-term artificial selection in Western breeds.

### 3.4. Tissue Specificity of SEs

To assess the genome-wide tissue-specific SE profiles, we identified SEs of eleven tissues (cerebellum, cerebrum, fat, heart, kidney, liver, lung, muscle, pancreas, spleen, thymus) with H3K27ac ChIP-seq data of Large White (cerebellum is used as an example in Figure 4A, and the results for other tissues are sorted in Appendix A). Pearson correlation analysis showed that the correlation between H3K27ac modification in different tissues was relatively low, with a coefficient between 0.29 and 0.83 (Appendix A), much lower than that of the same tissue between pig breeds (coefficients > 0.9). When comparing the SEs of the eleven tissues, we found a dramatic tissue specificity, with SEs being vastly diverse among tissues except for the lung, which appeared to share more SEs with fat and kidney (Figure 4B). Pearson correlation analysis further supported this observation. The upset Venn diagrams (Figure 4C) show that SEs exhibit higher tissue specificity than TEs (Figure 4D).

In addition, we overlapped SE-associated genes of eleven tissues and obtained 59 SE-associated tissue-conserved genes (Appendix A), with *RELA* and *SRSF2* presented as examples in Appendix A. GO analysis of these genes highlighted the essential physiological function of cells, such as negative regulation of the apoptotic process and positive regulation of the protein catabolic process (Appendix A). More importantly, we analyzed the SE-associated tissue-specific genes, finding they were closely related to the typical functions of each tissue. For example, the neuropeptide signaling pathway was highlighted in the cerebrum, myocardial contraction was highlighted in the heart, and skeletal muscle development and contraction were underlined in the muscle (Figure 4E).

### 3.5. Species Specificity of SEs

To investigate the functional specificity and conservation of SEs among species, we first identified hepatic TEs and SEs in pigs (LW), humans, and mice, respectively (Figure 5A, Appendix A). The genomic distribution showed that most SEs were located in the promoter regions (88.13%, 90.22% and 70.48%) in all three species (Figure 5B). Then, we compared SEs and TEs across pigs, humans, and mice using UCSC LiftOver (minMatch = 0.5). The results showed higher sequence conservation (96.47% of SEs and 89.03% of TEs) and additional function conservation (23.19% of SEs and 12.74% of TEs) between pigs and humans than between mice and humans (Figure 5C). Finally, genes associated with TEs and SEs among the three species were compared. As shown in Figure 5D, the proportion of orthologous genes related to TEs was larger than that associated with SEs, suggesting that the SEs contained more significant specificity in different species. Such high specificity of SE is consistent with our observations among various tissues (Figure 4C,D).

With respect to the potential functions, GO enrichment using SE-associated genes suggested that the top terms, including positive regulation of transcription from RNA polymerase II promoter and positive regulation of transcription, were similar among the three species (Appendix A). However, motif analysis indicated that the top SE-associated transcription factors differed among the species (Figure 5E). These results imply that SEs might be involved in the transcriptional regulation of crucial pathways via varied regulatory mechanisms among species. We then performed KEGG analysis using 118 conserved SE-associated genes across the three species (Appendix A). The terms closely related to liver function, such as alcoholic liver disease, fat digestion and absorption, and cholesterol metabolism, were all enriched (Figure 5F). However, GO analysis using species-specific SE-associated genes highlighted various detailed liver functions, again suggesting a significant role of SEs in cell-specific processes. Even so, SEs of different species showed diverse regulatory involvement. In pigs, SE-associated genes favored immune regulation, such as the significantly enriched terms of innate immune response and positive regulation of interleukin-8 production. The human-specific genes were closely related to detoxification functions, such as the detoxification of copper ions. In comparison, the SE-associated genes in mice tended to have functions related to glucuronidation and metabolism (Figure 5G).

## 4. Discussion

### 4.1. SEs Are Tissue-Specific ‘Network Hubs’ of *cis*-Elements

The *cis*-regulatory elements play a vital role in the selective expression of genes during development, environmental response, and diseases. Before the concept of ‘super enhancers’ was widely used, researchers noticed that cell identity and disease genes tend to be regulated by complex enhancer networks [11]. Our study made a comprehensive study of three kinds of *cis*-elements across different tissues, breeds, and species, and provides reliable evidence that SEs work as tissue-specific ‘network hubs’ of *cis*-elements. First, SEs overlap with other *cis*-elements with stronger transcription capacity. According to our identification strategy, SEs are generated based on H3K27ac ChIP-seq regions, which are not oriented to promoter regions. However, our results revealed that SEs were most enriched in the proximal promoter region, and contained the densest TF binding sites and QTL sites. This observation suggests that SEs are at the center of *cis*-regulatory networks. Second, the tissue specificity of SEs overpowered the difference between breeds and even species. The comparison among different breeds, species, and tissues hints that TEs are much more related to tissue specificity than species, while SEs are even more salient at this point. In other words, SEs are highly tissue-conserved based on their participation in tissue-specific functional genes and pathways. The fact that the similarities between humans and pigs are more significant than those between humans and mice is reasonable, due to the more similar organic physiological characteristics between humans and pigs [35], particularly in liver functions such as metabolism, digestion, and detoxification [36]. These results agree both with diverse SEs profiles across cell lines [3] and with the remodeling of SEs during differentiation [26].

### 4.2. The Promising Application of SE in Genomic Selection Breeding

The most significant difference between the Chinese and Western breeds we selected was in the intensity of selection based on the market demand. As mentioned above, the profiles of SEs are conserved based on cell/tissue function rather than breed or species; the different distribution of SEs between Chinese and Western pigs reflects the effect of selection based on the aspect of the transcriptional regulatory network. Interestingly, our study reveals three significant characteristic differences between Chinese and Western breeds.

(1) Western breeds have more TEs and stronger TE conservation, while having fewer SEs compared to Chinese breeds. This observation hints that the SEs have been ‘concentrated’ during the decades of breeding; modern breeding strategies based on quantitative genetics may screen out less powerful *cis*-elements regarding commercial traits at the single nucleotide level [37]. (2) The SEs of Western breeds are particularly associated with essential growth control genes, while TEs and promoters do not show such features. The standout secreted hormones *GH1* and *PRL* and their downstream transcriptional activators *STAT5A* and *STAT5B* first attracted our attention, as both *GH1* and *PRL* stimulate the growth of the liver and other tissues by promoting cell proliferation and angiogenesis [38]. More interestingly, *IGF1*, which is mainly secreted in the liver and stimulated by *GH1* [39], was highlighted as well. The *IGF1* promoter has recently been revealed to regulate porcine liver growth through long-distance regulatory action and to interact with regions around transcription start sites, CpG sites, and functional QTLs [40], confirming the critical *cis*-regulatory functions of SEs from another perspective. On the contrary, the genes associated with TEs and promoters of Western breeds are not distinct from the Chinese breeds, which suggests that, as a *cis*-element, SEs are more sensitive when reflecting the specific cell functions. (3) The breed-specific SEs were associated with breeding objectives. Artificial selection for desirable traits can yield rapid change in a relatively short period. Benefiting from traditional methods and recent genomic selection over time, Western breeds showed a significant advance in economic traits such as growth rate, muscle mass, feed efficiency, etc. [41], with related QTLs all significantly enriched in the SE regions. All three observations imply that SEs can facilitate narrowing the detection region of the genomic selection strategy, which has long been restricted by the vast size of QTLs and the limited SNP density of commercial chips [42].

## 5. Conclusions

In summary, we identified and compared the genome-wide profiles of *cis*-regulatory elements, particularly SEs, in porcine livers of both Chinese and Western breeds, then explored the differences with regard to promoters, TEs, and SEs among breeds, tissues, and species. Our observations suggest that SEs are the key ‘network hubs’ of *cis*-elements, and are highly tissue-specific. The significant enrichment of transcriptional initiation regions, TF binding sites, and QTLs in SEs, along with the fact that intensely selected Western breeds include more concentrated and more functional SEs, hint at the promising application of SEs in genomic selection breeding.

## Figures and Tables

**Figure 1 biology-11-01631-f001:**
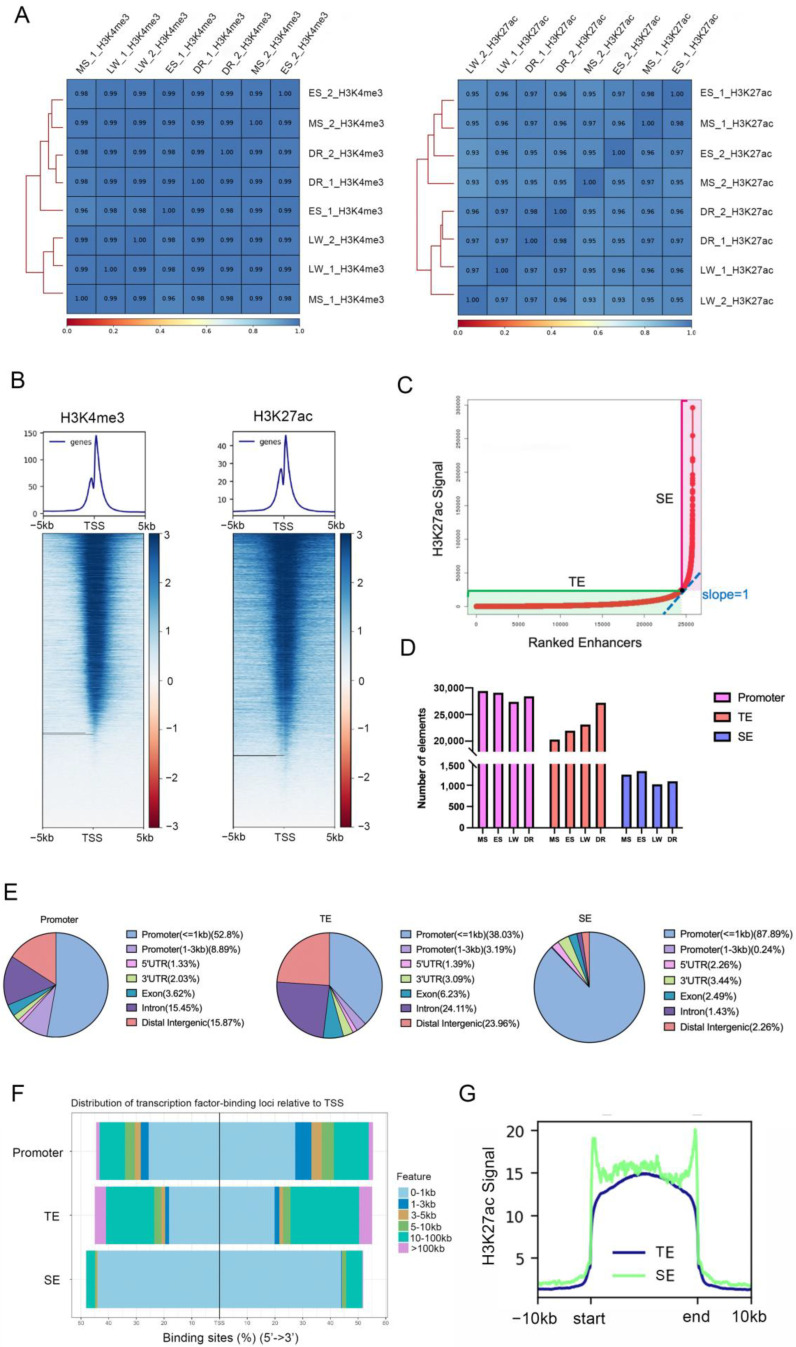
**The distribution and features of *cis*-regulatory elements in pig liver tissue.** (**A**) Correlation analysis among all samples for H3K4me3 and H3K27ac histone ChIP-seq. (**B**) Heatmaps depicting normalized ChIP-seq signal of H3K4me3 and H3K27ac at 5 kb near the TS, sorted by signal intensity. (**C**) The schematic for SE and TE identification. Promoters were identified by H3K4me3 signals, while TEs and SEs were identified by sorted H3K27ac using the ROSE algorithm. (**D**) The number of promoters, TEs, and SEs in porcine liver. We identified an average of 28,116 promoters, 20,295 TEs, and 1248 SEs. (**E**) Distribution of promoters, TEs, and SEs across the genomic regions. (**F**) Distribution of promoters, TEs, and SEs relative to TSS. (**G**) The plot shows average SE and TE signal levels in the porcine livers.

**Figure 2 biology-11-01631-f002:**
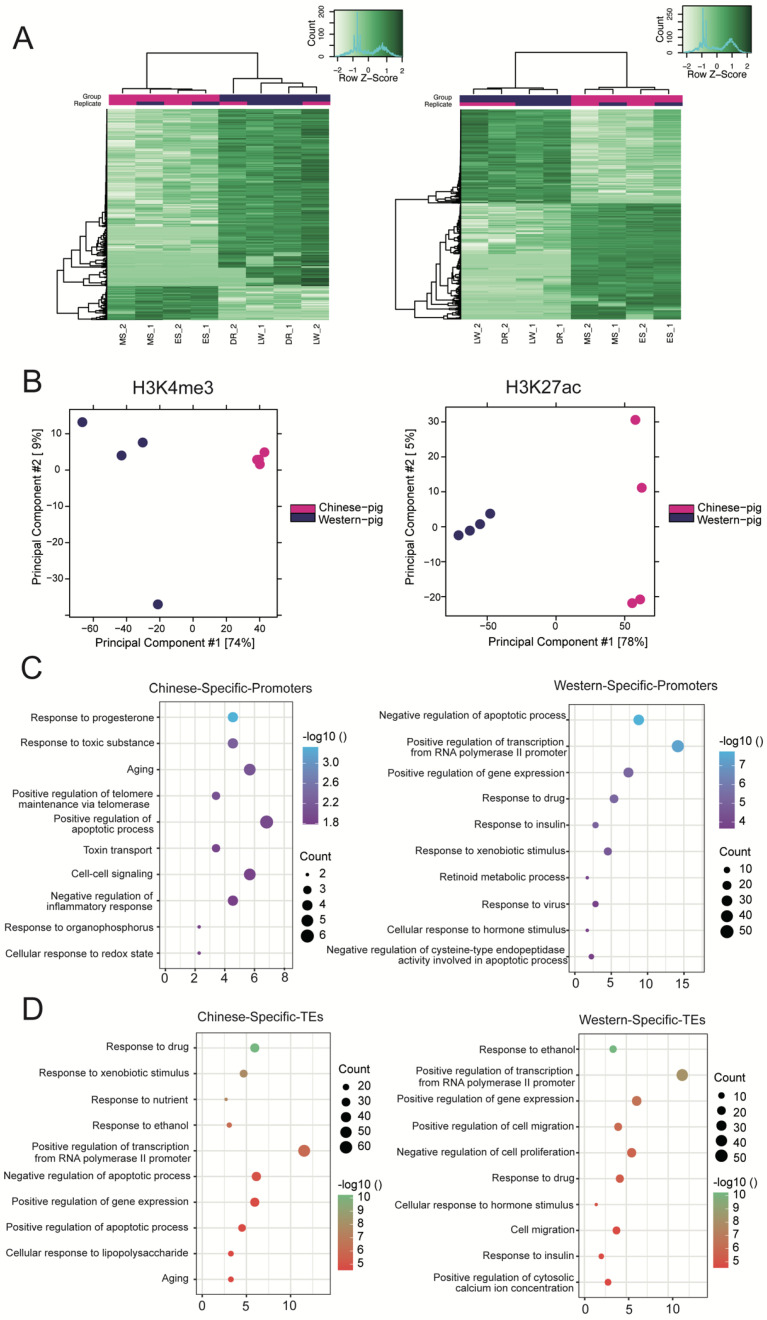
**Comparison of promoter and typical enhancer variants.** (**A**) Heatmap clustering of H3K4me3 (left) and H3K27ac (right) DERs across Chinese and Western breeds in porcine liver. (**B**) Principal component analysis (PCA) of H3K4me3 and H3K27ac DERs. (**C**) The GO terms of Chinese breeds and Western breed promoter DER-associated genes. (**D**) The GO terms of Chinese breed and Western breed TE DER-associated genes.

**Figure 3 biology-11-01631-f003:**
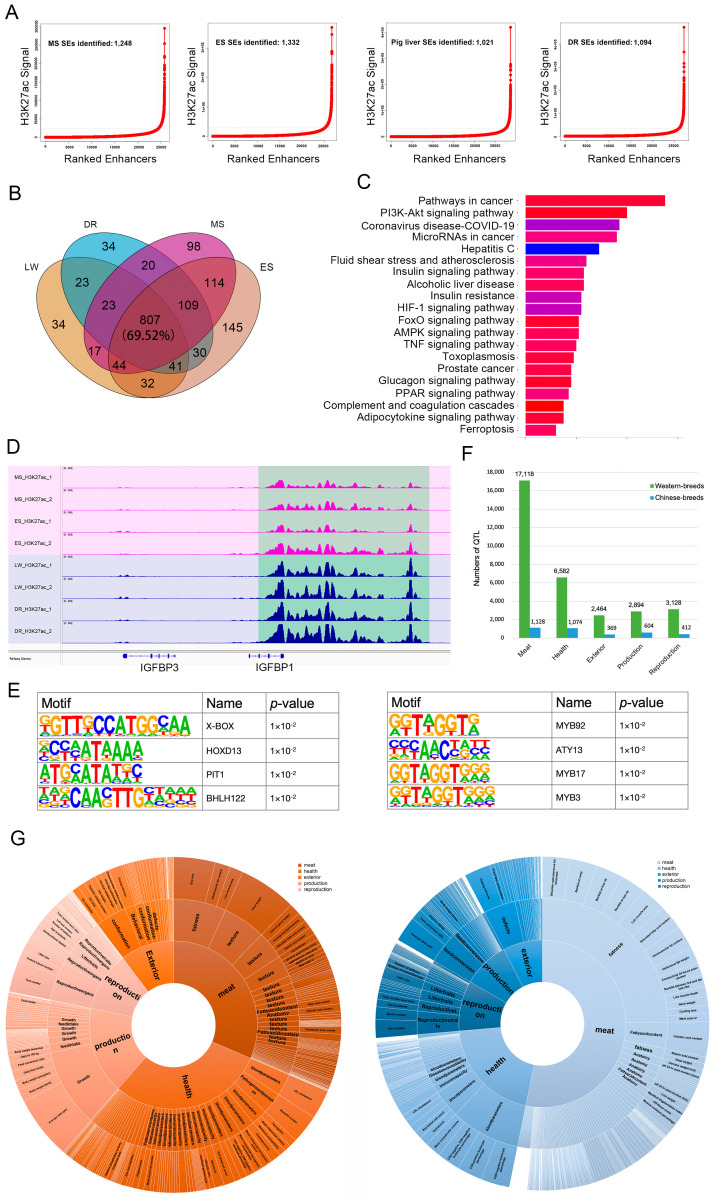
**Comparison of SEs between Chinese and Western breeds.** (**A**) Identification of SEs. Enhancers with a signal above a cutoff slope of one on the curve were considered SEs. (**B**) Venn diagram showing the number of breed-conserved SEs across four pig breeds. (**C**) KEGG pathway analysis of breed-conserved SE-associated genes. (**D**) The genome browser plot illustrates the regions with *IGFBP1* across four pig breeds. (**E**) Top TF motifs enriched in Chinese and Western breed-specific SEs. (**F**) The QTL number overlapped with Chinese and Western breed-specific SEs. (**G**) Distribution of the Chinese and Western breed-specific SE-associated QTLs.

**Figure 4 biology-11-01631-f004:**
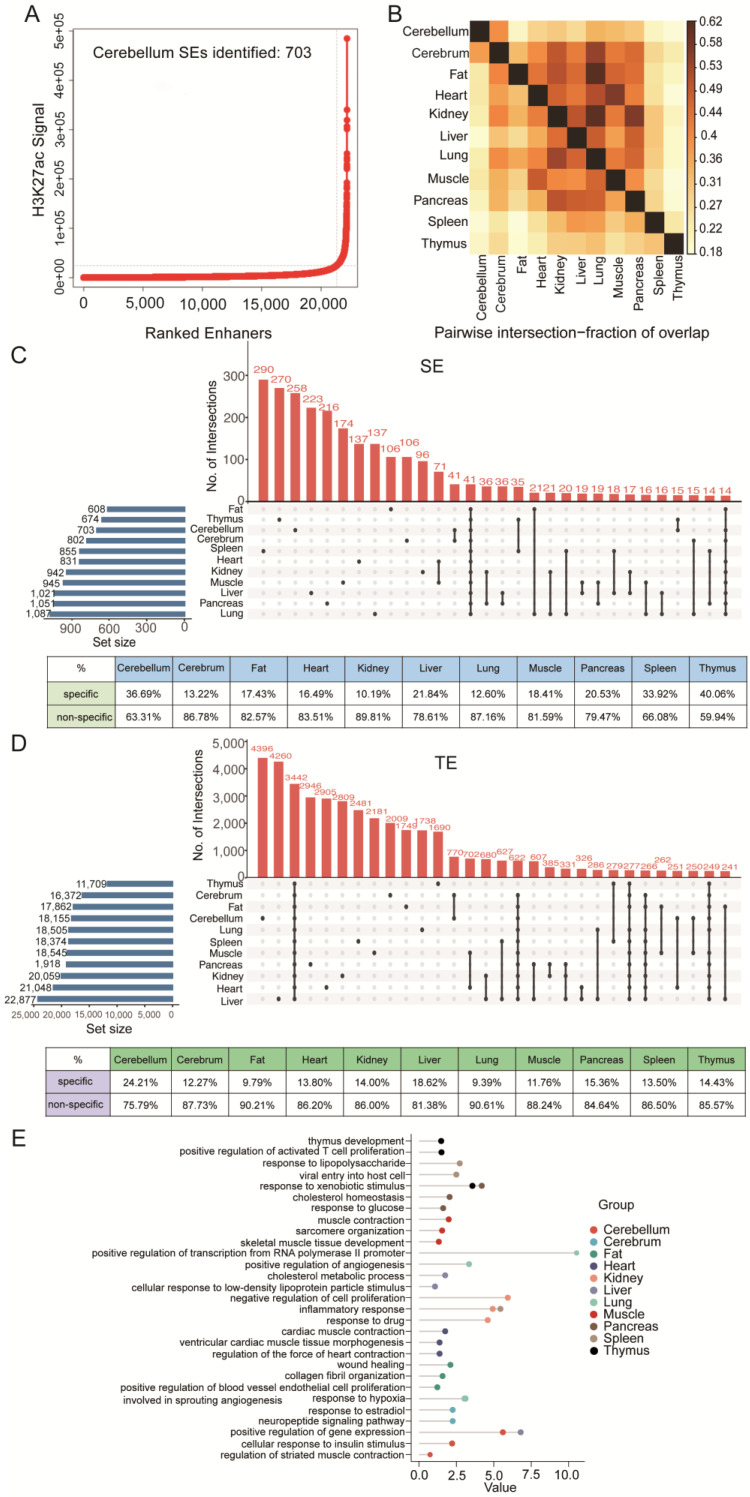
**Comparison of SEs among different tissues.** (**A**) Identified SEs in eleven tissues, using the cerebellum as an example. (**B**) SE correlation across the eleven tissues. (**C**) Upset Venn diagrams showing the overlap number of SEs in the eleven tissues. The color-coded tables show the percentages of specific and non-specific SEs for each tissue. (**D**) Upset Venn diagrams showing the overlap number of TEs in eleven tissues. The color-coded tables show the percentages of specific and non-specific TEs for each tissue. (**E**) The GO term of specific SEs in the eleven tissues. Each color represents one tissue, and the three most significant terms are shown in the figure.

**Figure 5 biology-11-01631-f005:**
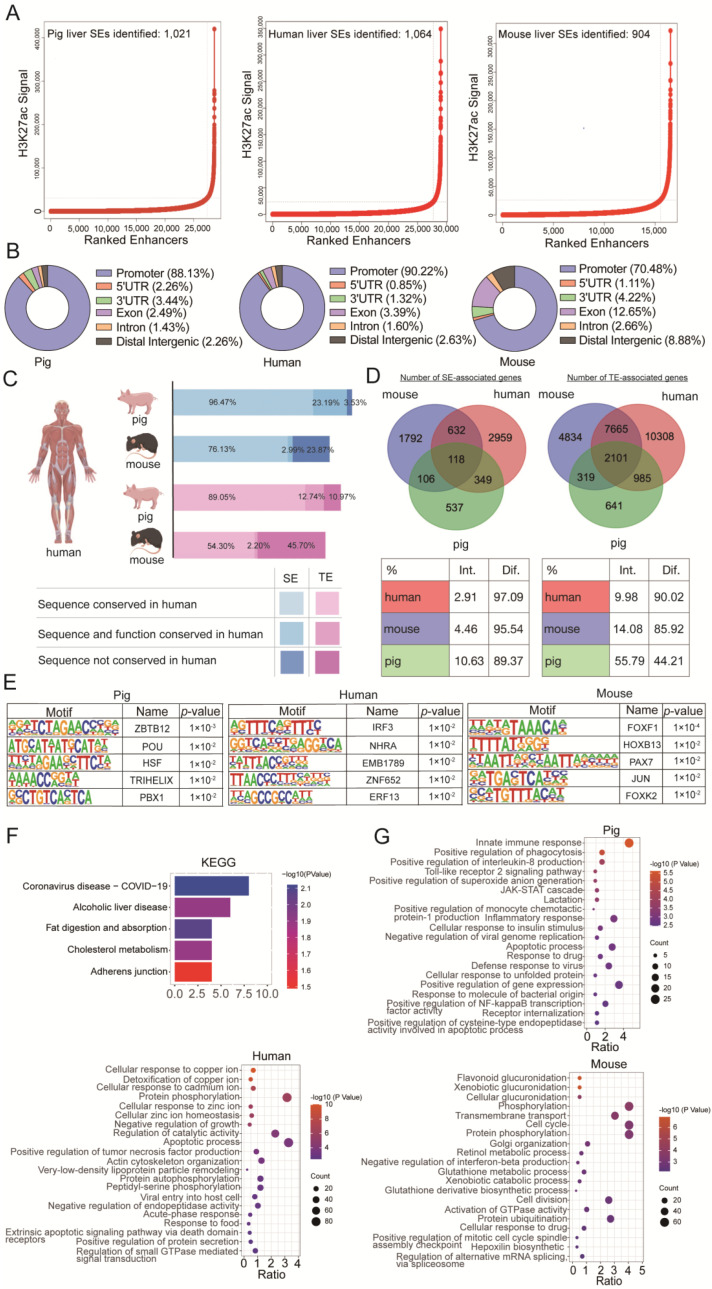
**Conservation of SEs across mammal species.** (**A**) Identification of SEs in pigs, humans, and mice. (**B**) Distribution of SEs across the genomic regions in pigs, humans, and mice. (**C**) Sequence and functional conservation of SEs and TEs between humans and other species. (**D**) Venn diagram of SE-associated liver genes (left) and TE-associated genes (right) in pigs (green), humans (red), and mice (purple). Color-coded tables show the percentages of intersection and difference gene numbers for each species. (**E**) Top TF motifs enriched in pigs, humans, and mice for specific SEs. (**F**) KEGG analysis of species-conserved SE-associated genes. (**G**) The GO terms of pigs, humans, and mice for specific SE-associated genes.

## Data Availability

All data are contained within the article.

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
