# Peer review of "Effective Quality Breeding Directions—Comparison and Conservative Analysis of Hepatic Super-Enhancers between Chinese and Western Pig Breeds"

_biology, 2022, doi:10.3390/biology11111631_

Round 1
Reviewer 1 Report
This study identifies the cis-regulatory elements in the livers of different pig breeds using different sources of data, and further analyzed the conservation of super-enhancers among different species. Moreover, they also reveal the features of regulatory elements located in the QTLs of pig economic traits. It is a great research work. There are some minor revisions.
1. Some introductions about the reason why the authors choose to identify the cis-regulatory elements in the livers can be added in the introduction section.
2. In 1. Introduction, Mutations of cis-elements which may synchronously… mutations ?
3. In section 2.3 and 2.4, add references for ChIpseeker (version 1.20.0)
4. As the presence of the results in section 3.3., the order of figure 3F and 3E should be exchanged in figure 3.
Reviewer 2 Report
Please see attached.
